# Probiotic Potential of Lactic Acid Bacteria Isolated from Moroccan Traditional Food Products

**DOI:** 10.3390/microorganisms13092201

**Published:** 2025-09-19

**Authors:** Ange Olivier Parfait Yao, Majid Mounir, Hary Razafindralambo, Philippe Jacques

**Affiliations:** 1BioEcoAgro Joint Research Unit, TERRA Teaching and Research Centre, Microbial Processes and Interactions, Gembloux Agro-Bio Tech, Université de Liège, Avenue de la Faculté 2B, bâtiment 140, 5030 Gembloux, Belgium; ange.yao@uliege.be (A.O.P.Y.); philippe.jacques@uliege.be (P.J.); 2Unité de Formation et Recherche (UFR) des Sciences Biologiques, Département de Biochimie-Génétique, Université Peleforo Gon Coulibaly, Korhogo BP 1328, Côte d’Ivoire; 3Food Technology and Bioengineering, Department of Food Science and Nutrition, Hassan II Institute of Agronomy and Veterinary Medicine, P.O. Box 6202, Madinat Al-Irfane, Rabat Instituts, Rabat 10112, Morocco; mounirmajid@gmail.com; 4Integrated and Urban Plant Pathology Laboratory, Gembloux Agro-Bio Tech, Liege University, Passage des Déportés 2, 5030 Gembloux, Belgium

**Keywords:** probiotic strains, lactic acid bacteria, antibacterial activity, MALDI-TOF/MS, 16S rRNA

## Abstract

This study assessed the performance and potential use of lactic acid bacteria (LAB) from Moroccan traditional foods as probiotics in animal feed. Five LAB strains *Lactiplantibacillus plantarum* from whey sourdough, *Leuconostoc pseudomesenteroides* and *Leuconostoc mesenteroides* from goat cheese, *Enterococcus durans* and *Lacticaseibacillus casei* from fermented milk were isolated and identified by 16S rRNA gene sequencing and MALDI-TOF mass spectrometry. Probiotic traits were evaluated by measuring acid/bile tolerance, cell surface hydrophobicity, emulsifying properties, antimicrobial activity and organic acid production, and safety checked through hemolysis and antibiotic sensitivity tests. *L. plantarum*, *L. casei*, and *E. durans* showed high survival rates after 24 h of culture under acid/bile stress conditions. The surface hydrophobicity of all strains ranged from 14.4 to 39.2%. *L. plantarum* showed the highest emulsifying capacity (81.4%) and stability (20%) after 24 h. Most strains inhibited pathogenic *Staphylococcus epidermidis*, *Bacillus cereus*, and *Escherichia coli*. Metabolite profiling revealed *L. pseudomesenteroides* as an interesting butyric acid-producing bacterium and *L. plantarum* as a remarkable strain releasing high content of organic acids. Their antibiotic susceptibility and non-hemolytic nature support their safety and potential use as feed additives.

## 1. Introduction

Growing environmental pressures and food safety issues are central concerns for scientists and society. It is no longer just a question of producing enough food, but also of ensuring that it is healthy, nutritious and free from contaminants [1]. A key challenge faced globally is the ability to satisfy rising demand for high-quality, nutrient-rich food while simultaneously safeguarding consumer health [2]. Simultaneously, the excessive and improper use of antibiotics as growth promoters or prophylactics in livestock has contributed to the emergence of antibiotic resistance in pathogenic microorganisms, including *Escherichia coli*, *Salmonella enteritidis*, *Staphylococcus aureus* and *Bacillus cereus* [3]. This resistance undermines the efficacy of antibiotics, posing a significant threat to both human and animal health [4].

Consequently, the European Union and USA implemented a prohibition on the uncontrolled use of antibiotic growth promoters (AGPs) [5], prompting increased interest in sustainable alternatives that benefit not only human health but also the environment, plants, and animals [6]. The reduction in antimicrobial usage in the agricultural sector presents a considerable opportunity, supporting a proposal to progressively eliminate their application as growth promoters [7,8]. Currently, extensive research endeavors are focused on identifying natural and sustainable solutions to mitigate the dual challenges of chemical residues and antibiotic resistance, thereby advancing the safety and sustainability of food and health systems [9]. Potential alternatives include the utilization of probiotics, which are live microorganisms that, when consumed in sufficient quantities, exert beneficial and regulatory effects on the microbiota, thereby influencing the host’s health positively [10]. The capacity of probiotics to enhance the balance of intestinal microbiota and barrier function, fortify the immune system, and contribute to the prevention of various diseases, is the subject of extensive global research. However, most studies predominantly focus on strains derived from products of European or Asian origin, leaving significant potential reservoirs of probiotics largely unexamined. Africa, with its remarkable cultural diversity and unique culinary traditions, represents a valuable yet underutilized source of probiotic strains [9,11,12]. Historically, these beneficial microorganisms have been isolated from healthy human hosts, animal, milk, and dairy products [12,13,14]. This concept has significantly advanced with the identification of novel probiotic strains in various fermented food and non-food products from diverse regions across the globe.

Africa is among the continents with the richest diversity of traditional fermented foods, which are of great importance regarding socio-economic growth, food security, nutrition, and health of African consumers [15]. In several African countries, fermented foods have been playing a pivotal role in food preservation and nutritional values, owing to the abundance of beneficial microbial populations with a wide range and multiple functionalities [16]. Nowadays, the progress in high-throughput DNA sequencing and proteomic technics for microbial identification can support the idea of screening food and non-food matrices from various African countries. In this context, our team has been focusing on potential material sources available in Morocco and in Western African countries for a collaborative work within the framework of the Horizon Europe URBANE project (www.urbane-project.eu/wp-urbane, accessed on 1 July 2022). The selection of these sources is guided by the Food and Agriculture Organization of the United Nations (FAO) and the World Health Organization (WHO) criteria, and fully in line with an approach to promote local biological resources and preserve a unique food and microbiological heritage, in accordance with the principles of sustainability and scientific sovereignty promoted under the Nagoya agreements [17]. In Morocco, as in West African countries, traditional fermented products are widely consumed by the population, especially in rural areas where the processes are still carried out according to ancestral practices transmitted locally. Due to the specific environmental conditions and the absence of industrialization, some food matrices in these regions likely contain endemic strains that have not yet been described. For instance, Moroccan cow milk often produced through back-slopping methods contains LAB strains with highly valuable properties and functionalities such as stress resistance and metabolite producing capacity [16].

The main purpose of this study was to identify and evaluate the LAB isolated from Moroccan food products (whey sourdough, goat cheese and fermented milk) by assessing their acid and bile tolerance, cell surface hydrophobicity, emulsifying capacity, antimicrobial activity, and production of short-chain fatty acids. The study also examines bacteria safety profiles, specifically their hemolytic activity and antibiotic susceptibility.

## 2. Materials and Methods

### 2.1. Sample Collection and Media Preparation

Samples were collected from three natural products (fermented milk, whey sourdough and goat cheese) provided by the Moroccan partner involved in the Horizon Europe URBANE project (https://urbane-project.eu/wp-urbane/, accessed on 1 July 2022). The raw materials were maintained at 4 °C during transport to the Faculty of Gembloux Agro-Bio Tech at the Terra Teaching and Research Centre in Belgium and were subsequently stored under the same conditions until further analysis. To isolate LAB strains based on their taxonomic group or genus, two selective culture media were used: De Man, Rogosa, and Sharpe (MRS) and M17 (Merck KGaA, 64271 Darmstadt, Germany) [18]. Media were prepared according to the manufacturer’s instructions. Calcium carbonate was added during the screening and isolation steps as an indicator of acid production. To avoid Maillard reactions during sterilization, glucose (20 g) was dissolved in 200 mL of distilled water and sterilized separately.

### 2.2. Isolation and Selection of LAB

One gram (1 g) of each sample of Moroccan natural products was diluted separately in 9 mL of sterile buffered peptone water. A series of successive dilutions was carried out and then spread on a Petri dish containing MRS or M17 agar medium following by incubation at 37 °C [18]. The LAB were isolated anaerobically in a jar [12] equipped with a deaerator (Oxoid^TM^ AnaeroGen^TM^ 2.5 L, Oxoid Ltd, Wade Road, Basingstoke, RG24 8PW, United Kingdom by Mitsubishi Gas Chemical Company Inc., Tokyo, Japan) to obtain their optimal growth parameters. Following a 24 to 48 h incubation period at 37 °C, colonies exhibiting phenotypic variations in terms of appearance, coloration, and morphology are selected. These colonies were subsequently subcultured onto the appropriate solid medium to obtain pure isolates, thereby ensuring accurate and reliable identification results [19,20].

### 2.3. Bacterial Species Identification

#### 16S rRNA Sequencing

Bacterial DNA was extracted using the NucleoSpin^®^ Tissue Kit (Macherey-Nagel) in accordance with the manufacturer’s protocol for bacterial samples. The purity of 2 µL of the extracted DNA was assessed using a Nanodrop spectrophotometer by measuring absorbance at 260 and 280 nanometers (nm), with 2 µL of distilled water serving as the blank. The samples were then purified using the NucleoSpin^®^ Gel and PCR Clean-up Kit (Macherey-Nagel, Düren, Germany). Subsequently, a polymerase chain reaction (PCR) was conducted utilizing Q5^®^ High-Fidelity DNA Polymerase. The primer sequences utilized were as follows: 8F (5′-AGA GTT TGA TCC TGG CTC AG-3′) and 1492R (5′-ACG GTT ACC TTG TTA CGA CTT-3′). The thermal cycling program employed consisted of cycles comprising 10 s at 98 °C, 30 s at 62 °C, and 1 min at 72 °C [21]. Amplification products were verified by agarose gel electrophoresis and subsequently submitted to Eurofins Genomics for sequencing. Bacterial identification was carried out by comparing the resulting sequences to those in the GenBank database using the National Center for Biotechnology Information (NCBI) BLASTn algorithm.

### 2.4. Maldi-TOF Mass Spectrometry Analysis

Bacterial isolates were identified using matrix-assisted laser desorption/ionization time-of-flight mass spectrometry (MALDI-TOF MS) model Bruker Autoflex Speed TOF/TOF (Bruker Daltonics, Bremen, Germany). Only isolates exhibiting distinct morphological characteristics were subjected to identification by this method. Individual colonies were first purified through successive streaking on agar medium to ensure clonal homogeneity. For protein extraction, approximately 1 mg of biomass was collected from a fresh colony using a sterile toothpick and suspended in 200 µL of ultrapure water. Subsequently, 600 µL of absolute ethanol was added to the suspension, which was then centrifuged at 13,000× *g* for 3 min. The resulting pellet was air-dried in a 40 °C oven for 5 min. Cell lysis was achieved by resuspending the dried pellet in 50 µL of formic acid (70%), followed by the addition of 50 µL of acetonitrile. Then, the suspension was centrifuged at 13,000× *g* for 2 min, and 1.25 µL of the supernatant was spotted onto a designated MALDI-TOF target plate. The plate was allowed to dry under a laminar flow hood. An equal volume (1.25 µL) of a saturated α-cyano-4-hydroxycinnamic acid (HCCA) matrix solution (Bruker Daltonics) was then applied to each well to facilitate ionization. After air-drying at room temperature, the plate was inserted into the MALDI-TOF instrument for spectral acquisition using FlexControl version 3.4 software. Mass spectra were recorded and analyzed using the manufacturer’s integrated software (FlexAnalysis version 3.4, 2021) and reference databases (MBT Compass reference library, 2021) [22]. Each isolate yielded a logarithmic score ranging from 0 to 3. According to the manufacturer’s guidelines, scores ≥ 2.300 indicate a highly probable species-level identification. Scores between 2.000 and 2.299 correspond to a secure genus-level identification and a probable species-level identification. Scores ranging from 1.700 to 1.999 are considered indicative of a probable genus-level identification, whereas scores < 1.700 are regarded as unreliable for taxonomic identification. Here, the identification is mainly based on the identification of ribosomal proteins that are very abundant in all microorganisms. The characteristic motifs of these proteins are compared with those stored in a large database, to determine the identity of the microorganism down to the species level [21].

### 2.5. Assessment of Resistance to Acid and Bile Salt

Acid and bile salt tolerance assays were conducted to evaluate the physiological robustness of the LAB isolates under gastrointestinal-like stress conditions. The assays were performed in MRS broth adjusted to pH 2.5 and 3.0 using 1 M sulfuric acid and supplemented with bile salts (Sigma-Aldrich Chemie GmbH, Steinheim, Germany) at final concentrations of 0.5% and 1.0% (*w*/*v*), in accordance with previously described protocols [23,24]. Bacterial growth under normal conditions (pH 6.5 without stress) was monitored over a 24 h period using a TECAN Spark^®^ multimode microplate reader. For this condition, 200 µL of the adjusted MRS broth standardized to an initial optical density (OD_600_) of 0.1 was aliquoted into individual wells of a sterile 96-well microplate. The microplates were incubated at 37 °C under anaerobic conditions, with continuous double orbital shaking (amplitude: 1 mm; frequency: 270 rpm). Optical density at 600 nm was automatically recorded every 20 min throughout the incubation period to assess bacterial growth kinetics under normal conditions.

After 4 h of culture in MRS broth supplemented with 0.5% or 1.0% (*w*/*v*) in acid conditions (pH 2.5 and 3) with a total volume of 10 mL in a flack of 100 mL, viability was determined by counting viable colonies on MRS agar incubated at 37 °C for 24 h in each stress condition and compared to the control (MRS pH 6.5), as described by Sakoui et al., 2024 [25].

### 2.6. Antimicrobial Activity

The antagonistic activity of the LAB isolates was assessed using an agar disk diffusion assay to evaluate their antibacterial potential. Briefly, a volume of 100 μL of each standardized indicator strains (10^6^ CFU·mL^−1^) was evenly spread onto Mueller–Hinton agar plates (Sigma-Aldrich, Darmstadt, Germany). Once the surface of the agar had dried, sterile Whatman paper disks were placed onto the medium, and 10 μL of each LAB young culture (10^8^ CFU·mL^−1^) were carefully deposited onto the disks and then incubated at 37 °C for 24 h. The selected indicator strains *Bacillus cereus*, *Staphylococcus aureus*, and *Escherichia coli* were obtained from the Terra Laboratory. Following 24 h of incubation at 37 °C, antibacterial activity was evidenced by the formation of clear inhibition zones (halos) around the disks. The diameter of each zone was measured in millimeters (mm) and interpreted as follows: no inhibition (<1 mm), weak inhibition (2–8 mm), moderate inhibition (8–16 mm), and strong inhibition (>16 mm) [26,27]. The results were divided as follows: no inhibition (<1), weak inhibition (2–8 mm), moderate inhibition (8–16 mm), and strong inhibition (>16 mm).

Antibacterial activity was also evaluated using the agar well diffusion method as mentioned by Arrioja-Bretón et al., 2020 [28]. In brief, indicator strains (*Escherichia coli*, *Bacillus cereus*, and *Staphylococcus epidermidis*) were cultured in nutrient broth to ~10^6^ CFU/mL and incorporated into molten Mueller–Hinton agar. Wells (8 mm) were aseptically punched and filled with 100 µL of cell-free supernatants (CFS) to measure only the influence of extracellular metabolites of LAB. Then, the supernatants were neutralized with 4 M NaOH to eliminate the influence of organic acids, acidification and hydrogen peroxides. Plates were incubated at 37 °C for 24 h, and antimicrobial activity was determined by measuring the diameter of the inhibition zones around the wells (mm). The neutralized cell-free supernatants (NCFS), which exhibited antagonistic effects against indicator strains, were considered as potentially producing bacteriocins or bacteriocin-like compounds. All experiments were performed in triplicate, and results are presented as mean ± SD (standard deviation).

### 2.7. pH Measurement

The pH of LAB cultures was measured to evaluate their acidifying capacity, a key criterion of probiotic potential and antimicrobial activity. The selected strains (*Lactobacillus plantarum* (E17), *Leuconostoc mesenteroides* (E19), *Leuconostoc pseudomesenteroides* (E18), *Enterococcus durans* (E21), and *Lactobacillus casei* (E22)) were grown in MRS broth at 37 °C for 24 h. Aliquots were collected at 0, 6, 12, and 24 h. pH was determined using a calibrated pH meter, standardized with buffer solutions at pH 4.0 and 7.0 prior to each measurement series. All measurements were performed in triplicate.

### 2.8. Surface Hydrophobicity Measurement

Surface hydrophobicity of isolated LAB was evaluated using the MATS (Microbial Adhesion To Solvent) method as originally described by Rosenberg et al., 1980 [29] subsequently modified by other researchers [30,31]. Toluene was employed as the hydrophobic solvent for this assessment. LAB were cultivated overnight in MRS broth devoid of Tween 80 (Sigma-Aldrich, Buchs, Suisse) to avoid interference of surfactants with solvent-water interfacial tension [30]. Cells were harvested at the late exponential phase and then centrifugated at 5000× *g* for 10 min at 4 °C, washed twice with sterile phosphate-buffered saline (PBS, pH 7.0), and resuspended in the same buffer. The optical density (OD_600_) of suspension was adjusted to 1.0 using a VWR-V 1200 spectrophotometer, corresponding to an approximate cell density of 10^8^ CFU·mL^−1^. For each assay, 1 mL of toluene was added to 3 mL of bacterial suspension in sterile glass tubes. The mixtures were vigorously vortexed for 3 min and then allowed to stand undisturbed at room temperature for 20 min to facilitate phase separation [30]. Following incubation, the aqueous phase was carefully collected, and its absorbance at 600 nm was measured. The extent of bacterial adhesion to the hydrocarbon phase was determined by comparing the OD_600_ values before and after mixing. Surface hydrophobicity was calculated using the following Equation (1).(1)$$Hydrophobicity\;\left(\%\right)=\left[\frac{\left({OD}_{600\;before}-{OD}_{600\;after}\right)}{{OD}_{600\;before}}\right]\times 100$$
**Equation (1):** Percentage of surface hydrophobicity of LAB strains [30].

### 2.9. Organic Acids Production

To quantify short-chain fatty acids (SCFAs) produced by the LAB strains, cultures were grown in MRS broth at 37 °C for 18 h. Following incubation, the cultures were centrifuged at 5000× *g* for 15 min at 4 °C to collect the cell-free supernatants. A volume of approximately 1 mL of each supernatant was then filtered through 0.22 μm pore-size membrane filters (Millipore, St. Louis, MO, USA) to remove residual particulates. The concentrations of organic acids were determined using high-performance liquid chromatography (HPLC; Agilent 1200 Series, Agilent Technologies, Santa Clara, CA, USA). Chromatographic separation was performed using a suitable column (Aminex^®^ HPX-87H 300 mm length × 7.8 mm diameter, Bio-Rad, Hercules, CA, USA) and a mobile phase consisting of 5 mM sulfuric acid at a flow rate of 0.6 mL/min. Detection was carried out using refractive index detector (RID). An injection volume of 15 μL was employed for each sample, with a total run time of 35 min per analysis. Quantification of organic acids was performed through peak area integration using Agilent OpenLab CDS version 2.3 software. The results were expressed in terms of concentration based on calibration curves generated from authenticated standards analyzed under identical chromatographic conditions. This analytical approach enabled the identification and quantification of major SCFAs, such as lactic, acetic, butyric, and malic acids, produced during fermentation, this allowed metabolic profiling of the LAB strains [32,33].

### 2.10. Emulsion Index (EI)

Emulsion index was determined according to the method described by Prasanna et al., 2012 [34], with modifications. A volume of 5 mL of cell-free supernatant was mixed with 5 mL of toluene in a sterile glass screw-cap tube (100 mm × 13 mm) and vortexed vigorously for 1 min to ensure thorough homogenization. The assay was performed in triplicate. The emulsifying activity was assessed by determining the emulsion index (EI), expressed as a percentage. Emulsion height was measured at less than 1 h (initial emulsion stability) and after 24 h (long-term stability), using the following Equation (2):(2)$$EI\;\left(\%\right)=\frac{he}{ht}\;\times \;100$$**Equation (2):** Emulsion Index measurement of culture supernatant.

Where *he* corresponds to the height of the emulsion and *ht* the total height of the sample and toluene mixture.

### 2.11. Antibiotic Susceptibility

Antibiotic susceptibility of the bacterial strains was evaluated using the disk diffusion method on Mueller–Hinton (MH) agar. An overnight culture in MRS broth was centrifuged, washed twice with sterile PBS, and adjusted to a final concentration of approximately 10^8^ CFU·mL^−1^. The bacterial suspension was then evenly spread onto the surface of MH agar plates. After drying, six antibiotic disks (Neo-Sensitabs™, ROSCO, Taastrup, Denmark) were aseptically placed on the agar using a sterile antibiotic disk dispenser. The plates were incubated anaerobically at 30 °C for 24 h. Antibiotic susceptibility was determined by measuring the diameter of the clear zones of inhibition surrounding the disks. According to the interpretive criteria established by the Clinical and Laboratory Standards Institute (CLSI), strains exhibiting an inhibition zone diameter of less than 15 mm were classified as resistant (R), whereas those with a diameter equal to or greater than 15 mm were considered susceptible (S) to the tested antibiotic [35].

### 2.12. Hemolytic Activity

Hemolytic activity was assessed using Columbia agar supplemented with 5% (*w*/*v*) defibrinated sheep blood [36,37]. Actively growing bacterial cultures were centrifuged and washed twice with sterile PBS (pH 7). The resulting pellets were resuspended in 6 mL of PBS to achieve an optical density of 1 at 600 nm. A 10 µL aliquot of each bacterial suspension was spotted onto the blood agar plates. *Bacillus subtilis* BBG258 control [38] was used as a positive control, while sterile PBS (pH 7) served as the negative control. Plates were incubated at 37 °C for 48 h under aerobic conditions. Following the incubation, hemolytic activity was assessed by observing the lysis of red blood cells in the medium surrounding the colonies. Hemolysis was classified into three categories: (i) α-hemolysis, indicated by a greenish discoloration around the colony (partial hemolysis); (ii) β-hemolysis, indicated by a clear, transparent halo (complete hemolysis); and (iii) γ-hemolysis, characterized by the absence of any visible zone (non-hemolytic). Strains displaying γ-hemolysis were considered non-hemolytic and therefore regarded as safe for potential probiotic use [39]. For optimal visualization, hemolytic reactions were assessed by positioning the blood agar plates against a transmitted light source, allowing accurate differentiation of hemolysis types.

### 2.13. Statistical Analysis

Data preprocessing and analysis were performed, respectively, using Microsoft Excel 365^®^ and GraphPad Prism^®^ version 10.5.0 Software (Boston, MA, USA). All experiments were conducted in triplicate. Results are presented as mean values of the three replicates and one-way analysis of variance (ANOVA) was performed following by post hoc multiple comparison Tukey test. Differences were considered statistically significant when the decision threshold was set at *p*-value < 0.05.

## 3. Results

The characterization strategy described by Figure 1 follows a systematic, multi-step approach to comprehensively evaluate potential probiotic candidates, integrating both phenotypic and functional analyses. Initially, bacterial strains are cultivated under solid and liquid culture conditions to assess growth dynamics and purity. Strains were identified using molecular (e.g., 16S rRNA sequencing) and analytic (Maldi-TOF MS) techniques to confirm taxonomic status. The isolates with high score and percentage of identity were selected then tested for resistance to acid and bile salts to simulate gastrointestinal survival, an essential probiotic parameter. Functional screening includes assays for antibacterial activity against pathogens (e.g., by agar diffusion) and cell surface hydrophobicity measurements to infer the potential for adhesion to host epithelial cells. Further safety and efficacy assessments involve hemolytic activity tests (to exclude pathogenicity), antibiotic susceptibility profiling to ensure resistance patterns align with regulatory guidelines. The study of the hydrolase activity of bile salts related to cholesterol metabolism is highlighted by the evaluation of the emulsion index. Finally, the production of short-chain fatty acids (SCFAs) was quantified, as these metabolites contribute to gut health. This multi-layered strategy ensures a holistic assessment of the viability, safety and functionality of probiotics, in line with FAO/WHO standards for the characterization of probiotics [40].

### 3.1. Screening and Identification of the LAB Strains

The results of the identification indicate a diverse array of bacterial species present in both food and non-food products originating from Africa (Table 1). Five LAB isolates belonging to four genera were formally identified through both 16S rRNA gene fragment sequencing and MALDI-TOF mass spectrometry. The results indicate that the percentage identities for 16S rRNA identification range from 97% to 100%, while the scores obtained from MALDI-TOF MS fall between 2.02 and 2.37. The LAB identified are predominantly derived from fermented products, notably sourdough, fresh goat cheese, and fermented milk. Only five representative LAB species identified with the highest scores (≥2.000) and percentages of identity and showing the same bacterial species name on both sides were selected for further analysis.

### 3.2. Resistance to Acid and Bile Salt

The growth profiles in normal (pH 6.5 and 0% bile salt) and under stress conditions (0.5% and 1% bile salt) are illustrated in Figure 2. The LAB strains showed distinct growth profiles in MRS broth at pH 6.5 without bile salt (Figure 2A). Among all isolates, *Enterococcus durans* was the only strain capable of sustaining growth under simulated gastrointestinal conditions (Figure 2B). Moreover, *E. durans* exhibited a growth rate of 0.19 h^−1^; at pH 3.0 supplemented with 0.5% bile salts, compared to 0.07 h^−1^ at pH 2.5 and 0.5% bile salts (Figure 3).

Figure 4 shows the viability of the LAB strains determined after 24 h under normal (control) and stress conditions (pH 3 and 0.5% BS) in MRS broth. Three LAB strains (*L. plantarum*; *E. durans* and *L. casei*) maintained a relatively high survival rate under 0.5% of bile exposure, yet significant inter-strain differences were observed (*p* < 0.05). *Enterococcus durans* E21 displayed the highest resilience, retaining more than 80% viability even at 0.5% at bile salts at pH 3, thus confirming its robust gastrointestinal survival potential. *Lactiplantibacillus plantarum* E17 and *L. casei* E22 exhibited intermediate tolerance, with survival rates ranging between 70 and 75%. In contrast, *L. pseudomesenteroides* E18 and *L. mesenteroides* E19 demonstrated higher sensitivity to these stress conditions.

### 3.3. Surface Hydrophobicity

Figure 5 presents the surface hydrophobicity (%) of the LAB strains as determined by the MATS (Microbial Adhesion to Solvent) assay using toluene as the hydrophobic phase. The results reveal statistically significant differences in hydrophobicity among the tested strains. *Enterococcus durans* and *Lactiplantibacillus plantarum* exhibited the highest hydrophobicity values, at 39.2% and 36.8%, respectively, suggesting a strong potential for adherence to hydrophobic surfaces such as intestinal epithelial cells. In contrast, *Leuconostoc mesenteroides* demonstrated the lowest hydrophobicity, with a value of 14.4%, indicating a comparatively reduced surface adhesion capacity. These inter-strain variations reflect differential cell surface properties that may influence their probiotic efficacy, particularly in terms of mucosal colonization.

### 3.4. Emulsion Index

The emulsion index (EI) of each strain was assessed at two time points (t = 0 and t = 24 h). As shown in Figure 6, the EI significantly decreased after 24 h for most strains. Notably, *Leuconostoc mesenteroides* maintained a stable emulsion index over the 24 h period, indicating sustained emulsifying capacity. In contrast, a marked decline in EI was observed for the culture supernatants of *Leuconostoc pseudomesenteroides*, *Lactiplantibacillus plantarum*, and *Enterococcus durans*, suggesting a reduction in emulsion stability over time.

### 3.5. Antibacterial Activity

The evaluation of the antimicrobial activity of tested LAB strains revealed significant differences (*p* < 0.05) both between species and according to the target pathogen (Table 2). *L. mesenteroides* (E19) exhibited the highest inhibitory activity against *E. coli* (21.17 ± 0.36 mm), significantly outperforming the other LAB tested strains. Against *B. cereus*, *L. plantarum* (E17) displayed the widest inhibition zone (19.14 ± 1.30 mm), confirming its strong probiotic and antimicrobial potential. Regarding *S. epidermidis*, the highest inhibition was also observed for *L. plantarum* (19.11 ± 0.45 mm), followed by *L. mesenteroides* (16.71 ± 0.51 mm, c3). In contrast, L. pseudomesenteroides (E18) showed no activity against *E. coli* and *B. cereus*, and only limited inhibition against *S. epidermidis* (8.43 ± 0.51 mm). Similarly, *E. durans* (E21) and *L. casei* (E22) exerted moderate effects, with inhibition zones ranging from 9.50 ± 1.11 mm to 15.94 ± 0.25 mm depending on the pathogen. Antibacterial activity performed with neutralized cell-free supernatants showed no inhibition spectrum.

### 3.6. pH Measurement

To comprehensively assess the metabolic activity of the tested LAB strains, we characterized their acidification kinetics over 48 h (Figure 7). The pH change profile varied notably among the tested strains. *L. plantarum* (E17) exhibited the fastest and most pronounced acidification, reaching a stable plateau around pH 3.7 within 12 h. *L. casei* (E22) followed a similar, yet more gradual, trajectory, culminating at pH 4.2. *E. durans* (E21) showed intermediate acidification, stabilizing around pH 4.1, whereas *L. pseudomesenteroides* (E18) and *L. mesenteroides* (E19) displayed the lowest acidifying capacity, with pH remaining above 4.5, even after 24 h.

### 3.7. Organic Acid Production

Organic acid production by LAB species was evaluated using HPLC (Figure 8). Each species exhibited a distinct profile of organic acid production, both in terms of type and concentration. Succinic acid was produced exclusively by two strains, with *Leuconostoc mesenteroides* generating the highest concentration, approximately 7 g/L. *Lactiplantibacillus plantarum* demonstrated the greatest capacity for lactic and acetic acid production, with concentrations of approximately 14 g/L and 8 g/L, respectively. *Leuconostoc pseudomesenteroides* ranked second in lactic acid production and was the only strain capable of synthesizing butyric acid, reaching nearly 5 g/L. In contrast to the other species tested, *Enterococcus durans* did not produce detectable levels of acetic acid.

### 3.8. Antibiotic Susceptibility

Antibiotic susceptibility is assessed in this study using a range of common antibiotics. The results presented in Table 3 reveal that all the isolates were sensitive to erythromycin, chloramphenicol, tetracycline and penicillin, whereas *L. pseudomesenteroides*, *L. mesenteroides*, *E. durans*, and *L. casei* exhibit resistance to streptomycin and kanamycin.

### 3.9. Hemolytic Activity

The hemolytic activity of the five identified LAB is checked on blood agar compared to both negative (PBS) and positive (*Bacillus subtilis* BBG258) controls (Figure 9). The results clearly show that no clear zones are observed around all isolates, indicating that all species evaluated were not hemolytic.

## 4. Discussion

### 4.1. Identification

In this study, five native LAB isolates (*L. plantarum* (E17); *L. pseudomesenteroides* (E18); *L. mesenteroides* (E19); *E. durans* (E21); *L. casei* (E22)) were identified by the MALDI-TOF/MS technique and 16S rRNA sequencing before characterizing their probiotic properties. The selected strains cultivated on solid agar or in liquid medium were identified with scores between 2.02 and 2.37, which were like those obtained with LAB isolated and identified from fish [22]. 

### 4.2. Microbial Profiling

Tolerance to gastrointestinal conditions, especially low pH and the presence of bile salt, are recognized as a relevant criterion in the functional characterization of probiotic strains [41]. The results provide critical insights on growth profiles of LAB. Under pH 2.5 or 3.0 and bile salt concentrations of 0.5 or 1%, *E. durans* demonstrated a notably higher tolerance to acidic stress and bile salt exposure, especially at pH 3 with 0.5% BS, maintaining robust growth through 24 h, consistent with prior findings for *E. durans* F3 isolated from the gut fresh water fish *Catla catla*, which survived up to 2% bile salts and pH 3 with minimal viability loss over 6 h [42]. In contrast, *L. plantarum*, *L. pseudomesenteroides*, and *L. mesenteroides* showed limited growth across all conditions, indicating their higher sensitivity to low pH and bile salts. These results suggest strain-specific variations in acid and bile tolerance, which are critical parameters for probiotic selection, especially for strains intended to survive gastrointestinal transit. Several studies have highlighted the importance of acid and bile tolerance in probiotic functionality. For instance, *E. durans* has previously been reported as a resilient LAB strain capable of surviving gastrointestinal-like conditions, aligning with our results [42,43,44,45]. Similarly, the limited growth observed for *Leuconostoc* species under stress conditions is highlighted by (Gu et al., 2023), who showed the viability reduction in *Leuconostoc mesenteroides* under bile salt challenge [46]. These findings emphasize the need for detailed in vitro screening of candidate probiotics before their application in food or therapeutic contexts. Bacteria like *E. durans*, showing robust growth in low-pH and bile-rich environments, may serve as promising probiotics for functional foods or supplements designed to deliver viable bacteria to the intestines. Its specific growth rate (h^−1^) under simulated gastrointestinal tract conditions (pH 2.5 and 3.0; bile salts at 0.5% and 1%) showed significant variations (*p* < 0.05) in growth kinetics depending on the applied physiological stresses. The results demonstrated reduced growth at pH 2.5 compared to pH 3.0, reflecting the detrimental effect of high acidity on bacterial viability, likely due to cellular membrane degradation and metabolic enzyme inactivation [47]. The addition of 1% bile salts increased this inhibition, consistent with mechanisms described in lactobacilli where bile acids disrupt membrane integrity and induce oxidative stress [48,49]. However, the significant differences observed between treatment groups suggest that *E. durans* possesses partial intrinsic resistance, potentially mediated by efflux pumps or modifications in membrane lipid composition, as observed in *Lactiplantibacillus plantarum* LM1001, as found by Lee et al., 2024 [50].

In previous works, the stability of a specific strain of *E. durans* LAB18s has been shown under conditions resembling the intestinal environment, characterized by a pH 3 and the presence of 1% of bile salt [51]. Moreover, it is recognized that Enterococcus genera possess the ability to survive, compete, and adhere to host cells within the gastrointestinal tract (GIT) that is an essential characteristic for their effective application as probiotics. For this reason, Enterococcus strains have been used in the poultry sector due to their ability to produce bacteriocins (enterocins) that are active against *Clostridium perfringens* [44]. Strains of *Enterococcus durans* have also been shown to exhibit hydrolytic activity mediated by the enzyme N-acetyltransferase, which facilitates their growth in bile salt-enriched environments, a key probiotic trait associated with gastrointestinal resilience [47,48].

The comparative analysis of bile salt tolerance across the five LAB isolates reveals marked interspecific variability, underscoring the adaptive strategies of individual strains to gastrointestinal stress. *Enterococcus durans* (E21) exhibited the highest survival rates, maintaining viability above 80% even under 0.5% bile salt conditions, a feature that highlights its remarkable robustness and metabolic adaptability to the hostile gut environment. Such resilience is consistent with previous reports identifying Enterococcus species as strong candidates for colonization of the small intestine due to their intrinsic bile salt hydrolase activity. Conversely, Leuconostoc strains (*L. pseudomesenteroides* (E18) and *L. mesenteroides* (E19)) showed high sensitivity, with viability declining completely at the same bile concentrations, reflecting their ecological adaptation to dairy niches where bile stress is absent [52,53]. Interestingly, *Lactiplantibacillus plantarum* (E17) and *L. casei* (E22) demonstrated intermediate survival profiles, suggesting strain-dependent variation in membrane fatty [54] acid composition and stress-response gene regulation, both of which have been linked to enhanced probiotic functionality. The overall trend confirms that bile tolerance is not uniformly distributed across LAB species but rather strain-specific and isolates such as *E. durans* (E21) may therefore constitute promising candidates for probiotic formulations targeting intestinal persistence.

The capacity to inhibit pathogenic germs is an essential property sought in probiotics [54]. Among the five LAB strains tested, four (*L. plantarum*, *L. mesenteroides*, *E. durans* and *L. casei*) showed an inhibition zone with diameters varying between 9.6 and 21.2 mm around the pathogens *Escherichia coli*, *Staphylococcus epidermidis*, and *Bacillus cereus*. These pathogenic strains were chosen due to their well-documented roles in over 90% of foodborne illnesses [55] and their occurrence in animal pathology [26].

Our results reveal a striking strain- and pathogen-specific antibacterial activity among the tested LAB isolates. The strain *L. mesenteroides* (E19) exhibited the most potent inhibition against *E. coli*, while *L. plantarum* (E17) demonstrated superior activity against both *B. cereus* and *S. epidermidis*. Our results are in line with those reported by Koo et al., 2015 [52]. In that study, multiple strains of LAB isolated from ground beef products were evaluated, and only a single strain of *Leuconostoc mesenteroides* demonstrated enhanced antibacterial activity against *Escherichia coli*. *L. plantarum* is well recognized for its ability to synthesize an array of antimicrobial compounds beyond organic acids, including bacteriocins such as plantaricins and other bioactive peptides. On the other hand, the absence of inhibitory activity in *L. pseudomesenteroides* (E18) against *E. coli* and *B. cereus* underscores that not all LAB taxa are inherently antimicrobial highlighting the strain specificity of probiotic efficacy. This variability has been documented in several fermented food-derived LAB isolates, where only a subset exhibited significant antagonistic properties. Furthermore, among 72 LAB strains isolated by Otero et al., 2006 [56], only 2 strains of *Lactobacillus gasseri* (CRL1421 and CRL1412) were able to show antimicrobial activity against *Staphylococcus aureus* due to their high production of H_2_O_2_ and lactic acid. These results reflect the high functional diversity among LAB, often driven by differential production of organic acids and specialized antimicrobial metabolites [36]. This phenomenon aligns with recent studies highlighting that LAB isolates display variable antagonistic profiles depending on their source, strain identity, and target pathogen [57,58], underlining the necessity of characterizing each strain on a case-by-case basis. The strain *E. durans* (E21) showed antagonistic effects against *E. coli*, *B. cereus* and *S. epidermidis*, as reported by other authors, based on the inhibition zone diameters, which are larger by 5 to 53% according to the pathogens [50,51]. These recent studies have highlighted the antagonistic potential of *Enterococcus strains* against major foodborne pathogens, particularly *Listeria monocytogenes*, *Staphylococcus aureus* and *E. coli*, mainly through the production of enterocins and other antimicrobial metabolites.

The species *L. casei* (E22) inhibited the pathogen *E. coli* with 15.94 ± 0.25 mm inhibition diameter compared to those found in previous studies (between 10 ± 2 mm to 20 ± 2.5 mm) [59]. The antagonistic power seems to be common in LAB from fermented products as shown by previous studies [60,61]. Neutralized culture supernatants showed no inhibition, indicating that the inhibitory effect of our LAB isolates is primary attributable to acidification, mainly through the production of lactic and acetic acids, rather than bacteriocin secretion. In our study, the total disappearance of inhibition following neutralization strongly suggests an acid-dependent mechanism.

The antimicrobial efficacy of LAB is largely influenced by their metabolic activity, particularly the production of organic acids during this process. Understanding the relationship between acidification kinetics and pathogen inhibition is essential for evaluating their probiotic potential. In this study, we investigated the pH change profiles of several LAB strains and their link with inhibitory effects against common foodborne pathogens. Our results demonstrate that the acidification kinetics of LAB are closely associated with their antimicrobial activity. *L. plantarum* (E17) exhibited the most rapid and pronounced pH decline, reaching a stable plateau around pH 3.7 within 12 h, and displayed the largest inhibition zones against *B. cereus* and *S. epidermidis*. This observation confirms that rapid lactic acid production constitutes a primary mechanism for pathogen inhibition [20,62]. *L. casei* (E22), which followed a more gradual acidification trajectory culminating at pH 4.2, exhibited moderate inhibition zones, whereas *E. durans* (E21), with an intermediate final pH of 4.1, also displayed partial inhibitory activity. These findings suggest a dose-dependent relationship between pH reduction and antimicrobial effect, in agreement with Chang et al., 2021, who demonstrated that both the kinetics and magnitude of acidification directly influence the antimicrobial efficacy of LAB [63]. In contrast, *L. pseudomesenteroides* (E18), whose pH remained above 4.4 even after 24 h, was ineffective against *E. coli* and *B. cereus*, confirming that insufficient acidification strongly limits antimicrobial activity. Neutralization of the supernatants abolished the inhibitory effect, further supporting the notion that organic acid production constitutes the primary mechanism of action. Thus, the hierarchy observed in our inhibition assays directly reflects the fast acidification kinetics of strains showing the highest antimicrobial potential, whereas those with slower or limited acidification display reduced inhibitory activity according to the strain of pathogen. These results underscore the importance of considering both the kinetics and magnitude of pH reduction when evaluating the probiotic and antimicrobial potential of LAB.

### 4.3. Metabolic Profiling

Organic acids are among pathogen-inhibiting metabolites produced by probiotics, especially LAB [64]. The analysis of culture supernatants by HPLC identified nine different organic acids varying in their acid strengths [65] and produced in various concentrations by the five LAB strains. Our study was able to successfully identify a remarkable *L. plantarum* strain producing a mixture of five organic acids in significant concentrations, such as lactate, acetate, malate, citrate, and propionate, as also demonstrated by other research works [66,67]. This corroborates the antimicrobial efficacy of this strain, as shown in previous studies, on the susceptibility of certain fungi and Gram-negative pathogens to lactic acid producer LAB strains [68,69]. In fact, most of the antimicrobial activities of probiotic strains are associated with their ability to produce and release into their environment, organic acids, including short-chain fatty acids (lactic acid, acetic acid, citric acid, malic acid, oxalic acid, succinic acid, butyric acid, propionic acid, valeric acid) [60,67].

The ability of the LAB isolates to produce other metabolites of interest can also be evaluated indirectly by measuring the functional properties caused by the presence of such bioactive compounds. For instance, the presence of bio-emulsifiers or biosurfactants in the cell-free supernatants is identified by the emulsifying index (EI). This parameter is related to the capacity of the selected strains to produce bio-emulsifiers, capable of solubilizing the hydrophobic phase (e.g., oils, hydrocarbons, etc.), making this carbon source available to microorganisms. This is a useful screening indicator for the competition mechanism of nutrient against pathogens. The LAB supernatant EI was determined just after the emulsification (t ~ 0) (EI0) and after 24 h (EI24), indicating the emulsifying capacity and stability of the supernatant content, respectively. The difference in EI0 of the selected LAB indicates a variability in the capacity of the supernatant to disperse the oil phase in the continuous aqueous phase. However, EI24 is quite similar for all selected LAB, indicating an equivalent power of the different strains in producing emulsion stabilizers. The ratio EI24*100/EI0 is another indicator of emulsion stability that is correlated to the rate of emulsion destabilization.

### 4.4. Safety

In this work, safety has been assessed by testing the sensitivity to antibiotics and hemolysis for all LAB strains, even though whole-genome sequencing was necessary to potentially identify genes associated with certain Enterococcus [70].

The assessment of antibiotic susceptibility represents a critical safety criterion for LAB prior to their selection as potential probiotic strains [64]. In the present study, phenotypic profiling of the isolated LAB strains revealed distinct antibiotic susceptibility patterns. Except for *L. plantarum*, all the isolates exhibited pronounced sensitivity to erythromycin, with inhibition zones exceeding 28 mm; notably, *L. pseudomesenteroides* demonstrated an exceptionally large zone (39.3 mm) suggesting a low risk of resistance gene transfer for these antibiotics. Conversely, *L. plantarum*, *E. durans*, and *L*. *casei* exhibited intrinsic resistance to streptomycin (0 mm), while Leuconostoc strains showed moderate to high sensitivity. All the strains demonstrated strong sensitivity to chloramphenicol (>21 mm). This is consistent with the previous studies where 96.9% of the 65 LAB tested were highly susceptible to chloramphenicol [71]. Tetracycline exhibited the broadest inhibitory range (21.0–40.7 mm), with *L. pseudomesenteroides* again most susceptible. While tetracycline resistance genes (e.g., *tetM*, *tetW* and *tetL*) have been sporadically detected in LAB from fermented foods, the absence of phenotypic resistance in our isolates suggests either low gene prevalence or non-functional expression, corroborating recent phenotypic surveys [72]. Penicillin efficacy was uniformly high, particularly against *L. pseudomesenteroides* and *L. mesenteroides*, consistent with the well-documented intrinsic sensitivity of LAB to β-lactams due to peptidoglycan-targeted cell wall biosynthetic pathways [73,74]. Resistance to kanamycin (≤14 mm) was noted in *L. plantarum*, *E. durans*, and *L. casei*, a pattern widely reported as intrinsic, non-transmissible aminoglycoside resistance in LAB and deemed acceptable under EFSA criteria [75]. Our results are in line with those of previous studies, who observed the high resistance of LAB to kanamycin (32.3%) [71]. In summary, none of the strains displayed acquired or multidrug resistance. Their maintained susceptibility to relevant antibiotics erythromycin, chloramphenicol, tetracycline, and penicillin confirm their safety profile for food and feed applications. The observed resistance to aminoglycosides (Erythromycin and streptomycin), being intrinsic (with the resistance genes (aac(6′)-aph(2”), ant(6) and aph(3′)-IIIa) and non-acquired, poses no regulatory concern under current probiotic strain selection frameworks, as described by Wong et al., 2015 [73].

In accordance with the recommendations of the European Food Safety Authority (EFSA), the assessment of hemolytic activity is highly advised for bacterial strains intended for use in food products, regardless of their GRAS or QPS status [76]. In the present study, the hemolytic activity of five bacterial isolates was assessed using Columbia blood agar plates. None of the tested strains exhibited α-hemolytic or β-hemolytic activity but demonstrated γ-hemolytic behavior. The results of this work aligned with those reported in other investigations that suggest no hemolytic activities were found in eight *Lactiplantibacillus* species isolated from fermented millet-based alcoholic beverages [77]. The safety property of LAB isolated from fermented non-dairy food products in China was evaluated in other studies, which revealed no evidence of hemolytic activity for these bacteria [74]. The five strains isolated in this present study were, therefore, not involved in the lysis of erythrocytes as shown by several studies reporting that probiotics need to exhibit no hemolytic activity [71,78].

## 5. Conclusions

This study, which focused on the performance assessment of probiotic strains from Moroccan traditional foods, shows the potential of dairy fermented products widely consumed by the local population to contain beneficial microbes for diverse applications. Five isolated LAB strains doubly identified by 16S rRNA sequencing and MALDI-TOF techniques reveal potential probiotic properties, especially for their robustness to survive under bile and acid stress conditions (*L. plantarum*, *L. casei*, and *E. durans*), and for their capacity to release a high number of multiple organic acids (*L. plantarum*). Their safety is shown by the absence of hemolysis and resistance to common antibiotics. Such results support their potential use as additives for sustainable animal farming. Other strains of isolates from West African countries are now under investigation.

## Figures and Tables

**Figure 1 microorganisms-13-02201-f001:**
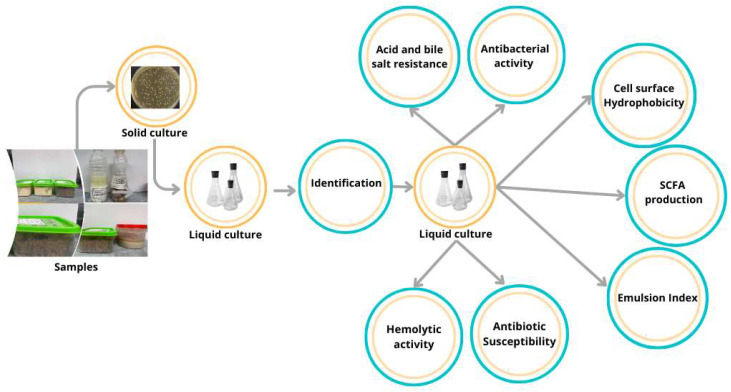
Mean steps for selection and characterization of potential probiotics.

**Figure 2 microorganisms-13-02201-f002:**
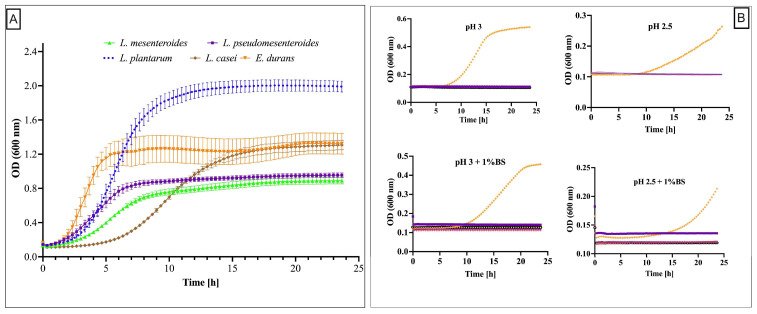
Growth profile over 24 h of LAB strains cultivated in (**A**) normal (pH 6.5 and 0% bile salt) and (**B**) under acid pH (top line) and acid pH with bile salt 1% (bottom line) stress conditions. Only *E. durans* grew under these stress conditions (orange line). The initial optical density was ±0.1.

**Figure 3 microorganisms-13-02201-f003:**
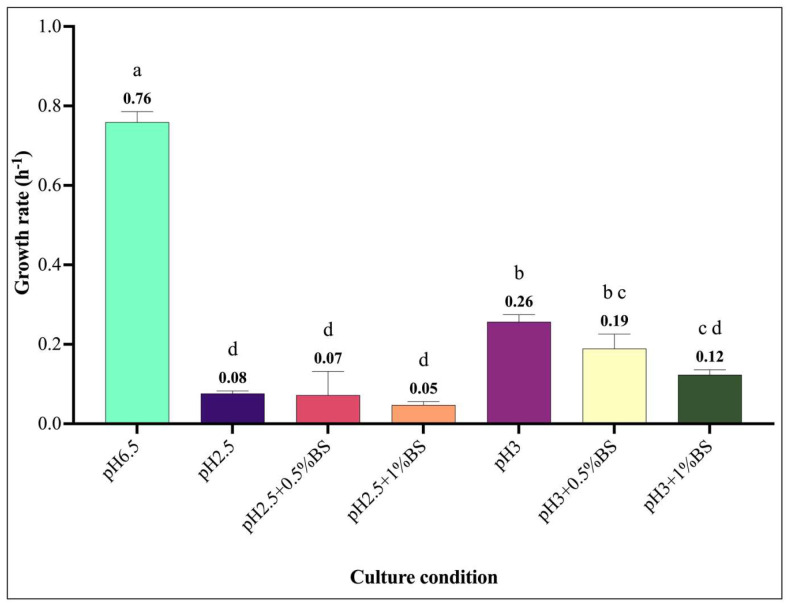
Growth rate (h^−1^) of *E. durans* in a medium simulating gastrointestinal tract condition at pH 2.5 and 3, with varying bile salt concentrations (0.5% and 1%) compared to control condition (pH 6.5). BS: Bile salt. Different lower-case letters indicated significantly different mean values ± SD (*n* = 3) (*p*-value < 0.05).

**Figure 4 microorganisms-13-02201-f004:**
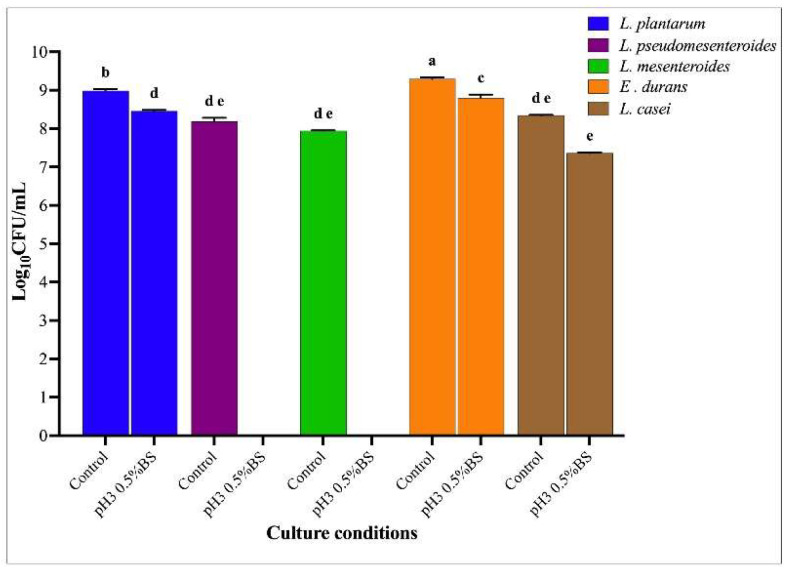
Viability (CFU/mL) of 5 LAB at different conditions (pH 3.0 and 0.5% BS and MRS broth pH 6.5 (control) after 24 h of culture. Different lower-case letters indicated significantly different mean values ± SD (*n* = 3) (*p*-value < 0.05).

**Figure 5 microorganisms-13-02201-f005:**
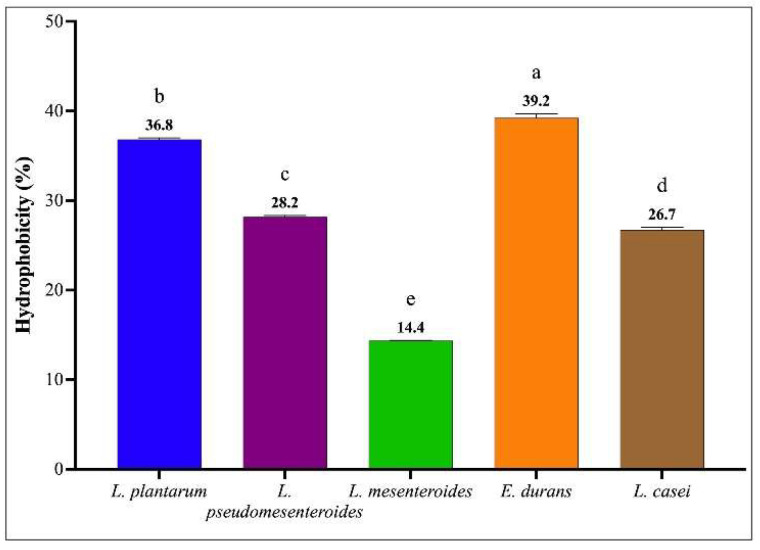
Surface hydrophobicity of LAB strains assessed by measuring the adhesion of PBS-washed cells (pH 7.0) to the hydrophobic solvent toluene (99% purity). Bars labeled with different lowercase letters indicate statistically significant differences in mean values ± SD (*n =* 3) among strains (*p* < 0.05), as determined by ANOVA one-way followed by post hoc Tukey test.

**Figure 6 microorganisms-13-02201-f006:**
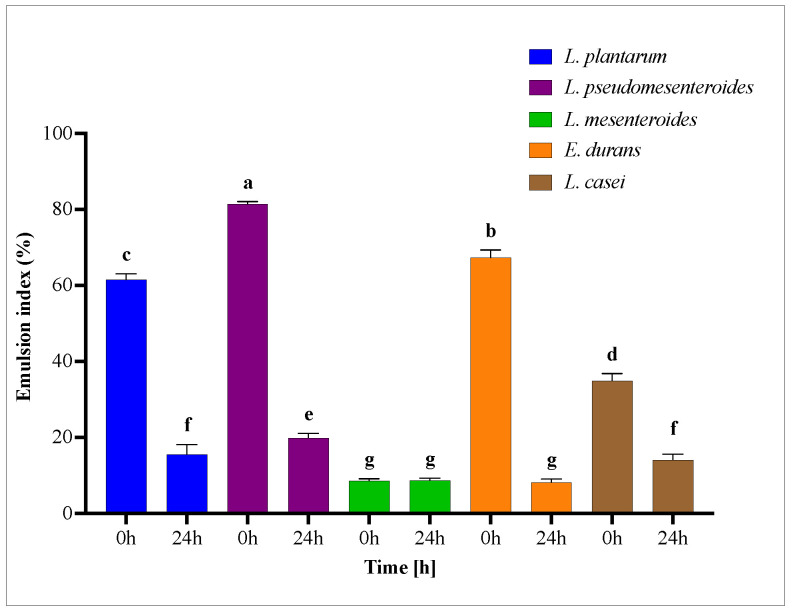
Emulsion Index at 0 h and after 24 h of cell-free supernatants in presence of toluene (*v*/*v*). Bars marked with different lowercase letters indicate statistically significant differences in mean values ± SD (*n* = 3) (*p* < 0.05).

**Figure 7 microorganisms-13-02201-f007:**
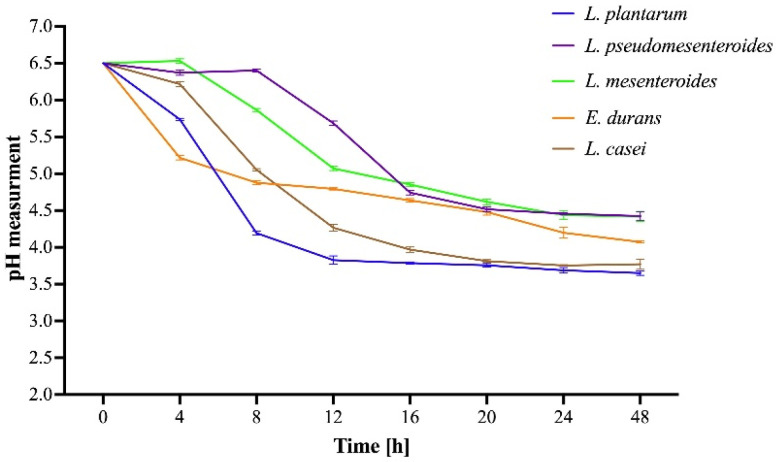
Monitoring of pH over 48 h for five LAB strains cultured in MRS broth and incubated at 37 °C with shaking at 150 rpm.

**Figure 8 microorganisms-13-02201-f008:**
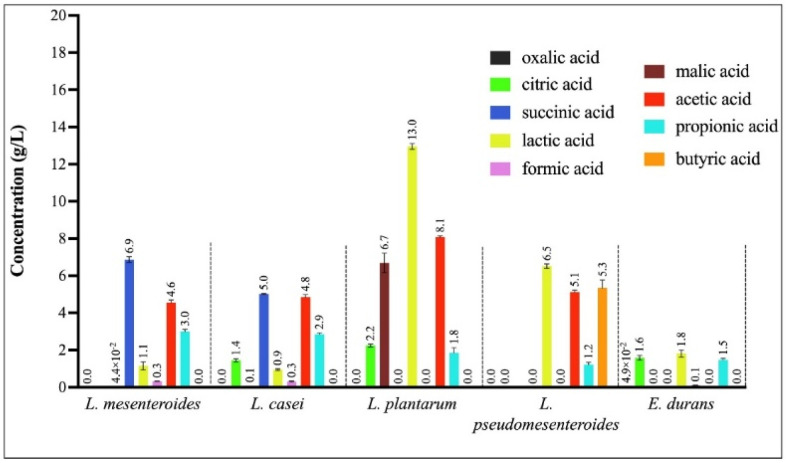
Qualitative and quantitative profiles of organic acids produced by the LAB strains.

**Figure 9 microorganisms-13-02201-f009:**
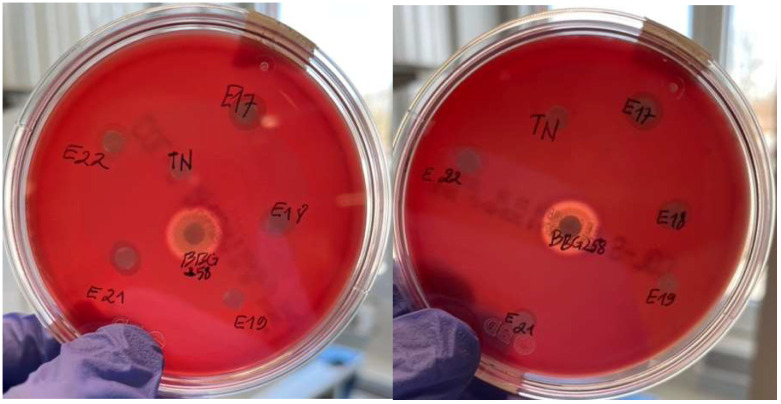
Hemolytic activity assay of five selected LAB compared with positive control (BBG258) and negative control TN (PBS at pH 7). (E17): *L. plantarum*; (E18): *L. pseudomesenteroides*; (E19): *L. mesenteroides*; (E21): *E. durans*; (E22): *L. casei*.

**Table 1 microorganisms-13-02201-t001:** List of isolates from food samples and identified species by MALDI-TOF/MS score and 16S rRNA identity percentage.

Matrice	ID	16S rRNA	Identity (%)	MALDI-TOF/MS	Score
Whey sourdough	E17	*Lactiplantibacillus plantarum*	97.38	*Lactiplantibacillus plantarum*	2.02
Goat cheese	E18	*Leuconostoc pseudomesenteroides*	98.48	*Leuconostoc pseudomesenteroides*	2.12
	E19	*Leuconostoc mesenteroides*	98.62	*Leuconostoc mesenteroides*	2.12
Fermented milk	E21	*Enterococcus durans*	96.62	*Enterococcus durans*	2.11
	E22	*Lacticaseibacillus casei*	100	*Lacticaseibacillus casei*	2.37

**Table 2 microorganisms-13-02201-t002:** Antimicrobial activities of the five LAB species against *Bacillus cereus*, *Staphylococcus epidermidis* and *E. coli* inoculated on MRS agar.

Species	Codes	*E. coli*	*B. cereus*	*S. epidermidis*
*L. plantarum*	E17	15.78 ± 0.69 (a_1_)	19.14 ± 1.30 (c_2_)	19.11 ± 0.45 (d_3_)
*L. pseudomesenteroides*	E18	0.00 ± 0.00 (c_1_)	0.00 ± 0.00 (d_2_)	8.43 ± 0.51 (a_3_)
*L. mesenteroides*	E19	21.17 ± 0.36 (b_1_)	15.03 ± 0.15 (b_2_)	16.71 ± 0.51 (c_3_)
*E. durans*	E21	15.30 ± 0.83 (a_1_)	10.34 ± 0.77 (a_2_)	9.60 ± 0.79 (a_3_)
*L. casei*	E22	15.94 ± 0.25 (a_1_)	9.50 ± 1.11 (a_2_)	10.47 ± 0.72 (b_3_)

Values represent mean inhibition zone diameters (mm) ± standard deviation (*n* = 3) measured from the edge of the disk to the margin of the clear zone. Anova one-way analysis followed by a post hoc Tukey Honest Significant Difference (HSD) multiple comparison test was performed for the three pathogens. Groups that share the same letter in each column are not different, but the difference is significant when the letters are different in the same column.

**Table 3 microorganisms-13-02201-t003:** Antibiotic susceptibility profiling of five LAB strains against six antibiotics.

Species	ERY [15 µg]	STR [10 µg]	CHL [30 µg]	TET [30 µg]	PEN [10 µg]	KAN [30 µg]
*L. plantarum*	22.0 ± 1.0 (d_1_)	0.0 ± 0.0 (a_2_)	21.3 ± 1.2 (d_3_)	21.0 ± 1.7 (c_4_)	28.3 ± 0.6 (b_5_)	0.0 ± 0.0 (c_6_)
*L. pseudomesenteroides*	39.3 ± 1.2 (e_1_)	25.7 ± 0.6 (c_2_)	36.0 ± 0.0 (e_3_)	40.7 ± 0.6 (b_4_)	41.0 ± 1.0 (d_5_)	25.3 ± 0.6 (d_6_)
*L. mesenteroides*	34.0 ± 0.0 (c_1_)	14.3 ± 0.6 (b_2_)	33.7 ± 0.6 (c_3_)	39.0 ± 1.7 (b_4_)	33.3 ± 1.2 (c_5_)	14.3 ± 0.6 (a_6_)
*E. durans*	31.3 ± 0.6 (a_1_)	0.0 ± 0.0 (a_2_)	24.7 ± 0.6 (a_3_)	32.3 ± 0.6 (a_4_)	19.7 ± 0.6 (a_5_)	14.0 ± 0.0 (a_6_)
*L. casei*	28.3 ± 0.6 (b_1_)	0.0 ± 0.0 (a_2_)	30.3 ± 0.6 (b_3_)	32.0 ± 1.7 (a_4_)	28.0 ± 2.0 (b_5_)	8.7 ± 1.1 (b_6_)

ERY: Erythromycin; STR: Streptomycin; CHL: Chloramphenicol; TET: Tetracycline: PEN: Penicillin; KAN: Kanamycin [µg]: concentration per disk of antibiotic (by agar diffusion method). Anova one-way analysis followed by a post hoc Tukey HSD multiple comparison test was performed for each antibiotic. Groups that share the same letter in each column are not different, but the difference is significant when the letters are different in the same column.

## Data Availability

The original contributions presented in this study are included in the article. Further inquiries can be directed to the corresponding author.

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
