# Peer review of "Probiotic Potential of Lactic Acid Bacteria Isolated from Moroccan Traditional Food Products"

_microorganisms, 2025, doi:10.3390/microorganisms13092201_

Round 1
Reviewer 1 Report
Comments and Suggestions for Authors
The article addresses an interesting topic: the selection and characterization of probiotics from African fermented products. The authors performed a series of tests including identification, gastrointestinal stress resistance, antagonistic activity, organic acid profiling, and safety evaluation (antibiotic resistance, hemolysis). However, the manuscript requires substantial revisions to meet the standards of Microorganisms.
- In my opinion, the abstract is too long and should be significantly shortened. Additionally, it includes non-scientific language such as “unequivocally identified” or “outstanding resistance”.
- In the introduction, I miss a clear explanation as to why these specific products were chosen for analysis and what novelty this study brings.
- There is a lack of detailed information regarding the MALDI-TOF parameters (e.g., software version, database used).
- The HPLC analysis should be better described, e.g., column parameters (dimensions) are missing.
- The methodology section lacks any mention of statistical analysis – including the software used, type of tests applied (e.g., ANOVA). P-values appear only in figure legends.
Below are some minor remarks:
- Line 22: “highlighting its robust for gastrointestinal resilience” – grammatical error; should be: “highlighting its robust gastrointestinal resilience”.
- Lines 36–37: “thereby, underscoring their strong potential…” – a better version would be: “thus underscoring their potential for application…”.
- Line 106: “was each diluted separately in 9 ml…” – grammatical error; should be: “was diluted separately in 9 ml…”.
- Line 137: “a small amount of biomass was collected…” – vague expression; better: “approximately 1 mg of biomass was collected…”.
- Line 141: “After a brief centrifugation…” – replace with: “centrifuged at 13,000 × g for 2 minutes”.
- Line 183: “others researchers” – grammatical error. Correct form: “other researchers”.
- Line 185: “to avoid interference of surfactants with interfacial tension” – better phrased as: “to avoid interference of surfactants with solvent–water interfacial tension”.
- Line 218: “thereby providing insight…” – a better version would be: “This allowed metabolic profiling of the LAB strains.”
- Line 347: “are summarized in” – incomplete sentence. Suggestion: “The inhibitory activities of the LAB species are summarized in Table 2.”
- Lines 390–411: The description of the MALDI-TOF identification procedure should be moved to the Methods section rather than being part of the Discussion.
- Lines 566–574: The conclusions are overly optimistic, e.g., “enhances their ability to competitively inhibit intestinal pathogens” – a more appropriate formulation would be: “suggests potential for competitive exclusion of intestinal pathogens, pending in vivo confirmation.”
Reviewer 2 Report
Comments and Suggestions for Authors
The article is interesting in that African food products are used as sources of probiotic strains. However, there are some questions regarding the article.
-
The authors list seven fermented products from which they isolated strains. Then, they mention five isolated strains. Please correlate which strains were isolated from which products. Provide data on the content of lactic acid bacteria in the various fermented products you used.
-
The authors present the antimicrobial activity of each strain, but it is unclear whether this activity was due to antimicrobial substances produced or to lactic acid production and acidification. In future studies, it is necessary to include two variants — with the native suspension and with the suspension adjusted to neutral pH. This will help to distinguish different types of antimicrobial activity.
-
In section 4.1, the authors describe the advantages of the MALDI-TOF/MS method at length. This is well-known information and can be significantly condensed.
Reviewer 3 Report
Comments and Suggestions for Authors
While the study valuably addresses an insufficiently studied Moroccan-derived LAB strain and provides foundational probiotic characterization data, it could be strengthened by addressing some methodological limitations, deepening the discussion, and refining the conclusions to more closely align with the findings. The impact of the work is somewhat constrained by the conventional experimental design, the need for more rigorous safety assessment (particularly for E. durans), and the limited connection drawn between the results and potential practical applications.
1. The isolated strains (L. plantarum, E. durans, etc.) are taxonomically common. The study did not identify novel species or strains exhibiting exceptional or previously unreported probiotic traits. Consequently, many of the conclusions regarding strain properties (e.g., E. durans stress tolerance) largely echo findings from prior studies.
2. Given that Enterococcus is a genus containing pathogenic strains, the safety assessment of E. durans requires more rigorous evaluation. The screening was limited to basic hemolysis and antibiotic susceptibility tests; crucial virulence genes (e.g., gelE, esp) were not investigated.
3. The bile salt concentrations tested (0.5–1%) fall significantly below physiological levels typically encountered in the mammalian small intestine (1–3%). Furthermore, relying solely on OD₆₀₀ measurements to assess viability, without complementary methods like CFU counts or live/dead staining to confirm membrane integrity and culturability, limits the robustness and physiological relevance of these results.
4. The potential contribution of organic acids present in the supernatants to the observed emulsification activity was not accounted for, as neutralization controls were omitted.
5. To clearly attribute antimicrobial activity to bacteriocins rather than acidification, controls such as pH adjustment or protease treatment of the supernatants were necessary but not performed.
6. Some seemingly contradictory results warrant further clarification. For instance: L. plantarum exhibited poor acid tolerance yet strong antimicrobial activity; E. durans reportedly lacked acetic acid production (which would typically be expected and could influence activity).
7. It would be beneficial to add error bars to all figures where applicable (e.g., Figure 6). Additionally, Tables 2 and 3 lack statistical analysis to indicate the significance of the differences presented.
Round 2
Reviewer 1 Report
Comments and Suggestions for Authors
All the necessary corrections have been made, and the responses provided by the authors are satisfactory. In my opinion, the manuscript now meets the standards Microorganisms and is suitable for publication in its current form.
Reviewer 2 Report
Comments and Suggestions for Authors
I have no any comments to authors
Reviewer 3 Report
Comments and Suggestions for Authors
The authors have carefully answered my questions and revised the manuscript.